# Bionanocellulose/Poly(Vinyl Alcohol) Composites Produced by In-Situ Method and Ex-Situ/Impregnation or Sterilization Methods

**DOI:** 10.3390/ma14216340

**Published:** 2021-10-23

**Authors:** Aldona Długa, Jolanta Kowalonek, Halina Kaczmarek

**Affiliations:** 1Bowil Biotech Sp. z.o.o., 7 Skandynawska St., 84-120 Władysławowo, Poland; dluga_aldona@wpl.pl; 2Faculty of Chemistry, Nicolaus Copernicus University in Toruń, 7 Gagarina St., 87-100 Toruń, Poland; halina@umk.pl

**Keywords:** bionanocellulose, poly(vinyl alcohol), bacterial synthesis, composite characterization

## Abstract

The purpose of the work was to obtain composites based on bionanocellulose (BNC) and poly(vinyl alcohol) (PVA) for specific biomedical and cosmetic applications and to determine how the method and conditions of their preparation affect their utility properties. Three different ways of manufacturing these composites (in-situ method and ex-situ methods combined with sterilization or impregnation) were presented. The structure and morphology of BNC/PVA composites were studied by ATR-FTIR spectroscopy and scanning microscopy (SEM, AFM). Surface properties were tested by contact angle measurements. The degree of crystallinity of the BNC fibrils was determined by means of the XRD method. The mechanical properties of the BNC/PVA films were examined using tensile tests and via the determination of their bursting strength. The water uptake of the obtained materials was determined through the gravimetric method. The results showed that PVA added to the nutrient medium caused an increase in biosynthesis yield. Moreover, an increase in base weight was observed in composites of all types due to the presence of PVA. The ex-situ composites revealed excellent water absorption capacity. The in-situ composites appeared to be the most durable and elastic materials.

## 1. Introduction

Cellulose is an excellent, hydrophilic, semi-crystalline biopolymer, an object of intensive scientific research, and a desirable raw material for applications in medicine and many industrial branches [1,2,3].

Although cellulose in its native form is not sufficiently versatile for direct processing, many chemical functionalization methods (e.g., esterification using organic or inorganic acids) can be used to create materials with specific thermoplastic properties [4]. Modified cellulose can form films or fibers of great practical importance. Because the number of cellulose solvents is limited [5], two fundamental processes for producing cellulose fibers are based on the usage of aggressive agents. Sodium hydroxide and toxic sodium disulfide are applied in the rayon process, while *N*-methylmorpholine *N*-oxide is applied in the Lyocell method. Great efforts are being made to eliminate these solvents from the production process. One possible solution is using ionic liquids, which are considered environmentally friendly due to their non-toxicity, non-flammability, extremely low vapor pressure, and good thermal stability [6].

Contemporary scientific research is focused on the modification of cellulose, aiming towards new applications, for instance, in medicine [3,7,8] (e.g., as a material for biomedical implants or scaffolds and dressings for difficult-to-heal wounds), pharmacy (as controlled release drug carriers), industry (as barriers, membranes, and absorbers), and electronics [9,10,11,12,13] (as conductive, magnetic, piezoelectric, or electromagnetic interference shielding materials). The use of cellulose as a matrix for nanocomposites that are of particular importance in modern nanotechnologies is also noteworthy. For example, the addition of silver nanowires to carboxymethyl cellulose (CMC) led to the obtaining of conductive composites that are useful in 3D-printing technology [14], and the introduction of carbon nanotubes (CNT) to nanocellulose caused very high conductivity (7.7 S/m) of the formed material [9]. The usage of cobalt ferrite (CoFe_2_O_4_) nanoparticles resulted in the creation of a composite with magnetic properties (with a magnetization value of ~13 emu/g) [11]. Combining metal–organic frameworks with cellulose nanofibers resulted in the formation of nanocomposites that were characterized by good thermal insulation and fire retardancy [15].

Moreover, cellulose fibers constitute the reinforcing phase of matrices when added to other polymers due to their high tensile strength and stiffness. Cellulose fibers successfully replaced glass fibers in composites owing to a significant improvement in their mechanical properties. Recently, a nanostructured material based on bacterial cellulose and graphene oxide, with excellent mechanical properties, was fabricated by the original method of layer-by-layer biosynthesis [16].

Cellulose can come from a variety of sources. The most common type is vegetable cellulose, e.g., from wood or cotton. The other type is algal cellulose that is present in the cell wall of algae. Due to their size and structure, the algae can be divided into microalgae and macroalgae [17,18]. Another type is cellulose of bacterial origin, i.e., bionanocellulose (abbreviated as BNC or BC; also called microbial cellulose, MC), produced by various strains of bacteria, e.g., *Gluconacetobacter xylinus, Rhizobium, Agrobacterium, Rhodobacter*, or *Sarcina*. The prefix “bio” in the name of the cellulose designates the latter type. Moreover, because of the nanometric diameter of the cellulose fibers, the prefix “nano” is also justified.

The main difference between these cellulosic materials is the form in which they occur and their chemical composition. In the case of the cellulose of bacterial origin, a three-dimensional structure of arranged nanofibrils is created (i.e., with dimensions much smaller than cellulose fibers of plant origin) [19]. The main advantage of BNC is its high purity as this material is free of any impurities, such as lignin, pectin, or hemicellulose, that are usually present in vegetable cellulose. BNC, as a material obtained from renewable sources, is gaining more and more interest due to its unique properties (high purity, biocompatibility, and biodegradability) and well-controlled synthesis [19,20,21,22,23]. Moreover, BNC is characterized by a high degree of crystallinity but relatively poor elasticity.

Recently, the increase in BNC applications has been observed due to the various possibilities for modifications or for the production of composites based on this polysaccharide [19,24,25]. BNC with biological activity can be obtained by mixing it with silver nanoparticles. Such material exhibited good antibacterial activity against both Gramm-negative and Gram-positive bacteria (*Escherichia coli* and *Staphylococcus aureus*) [26]. Modification of BNC by hydroxyapatite resulted in the formation of valuable material for bone tissue engineering [27]. Attractive, innovative materials have been obtained by combining BNC with other biopolymers such as collagen [28], starch [29], chitosan [30], and gelatin [31].

The main objective of the present work was to obtain and characterize the composite, made of bionanocellulose and poly(vinyl alcohol), designed for biomedical and cosmetic applications. Three methods of composite preparation were elaborated: in-situ—a direct culture of bacteria on PVA-modified culture medium; and two ex-situ processes—using the impregnation or sterilization of previously obtained BNC. The introduction of PVA into cellulose aimed at improving the mechanical properties and water absorption of this material, which was designed for wound healing. This synthetic polymer was selected because of its water-solubility, hemocompatibility, non-toxicity, and compatibility with polysaccharides due to the presence of hydroxyl groups that are capable of hydrogen bond formation. Although some publications describe BNC and PVA blends [32,33,34,35,36], there is still insufficient knowledge about these materials in terms of their broader applications. Our work included more comprehensive research on the preparation methodologies of BNC/PVA composites and their characterization by means of complementary instrumental techniques, with the intention of proposing specific uses.

## 2. Materials and Methods

### 2.1. Materials

PVA, with an average molecular weight of 30,000–70,000 and a hydrolysis degree of 87–89%, was purchased from Aldrich. The owner of the bacterial strain used for BNC synthesis—the *Gluconacetobacter xylinus* E25 strain—is Bowil Biotech Sp. z.o.o. (Władysławowo, Poland), where bacterial cultures were produced under controlled laboratory conditions. Culture medium components—D-glucose, Na_2_HPO_4_, MgSO_4_·7H_2_O, C_6_H_8_O_7_·H_2_O, NaOH, CH_3_COOH—were supplied by STANDLAB Ltd. (Lublin, Poland), and Yeast Extract was supplied by BTL Ltd. (Zakład Enzymów i Peptonów, Łódź, Poland).

### 2.2. BNC Synthesis

The production of bionanocellulosic material consists of two main stages: submerged culture fermentation and a stationary step. The culture medium was Schramm Hestrin (SH) or modified SH medium (in the case of the BNC/PVA composites). The first fermentation stage involved preparing the producer of the *Gluconacetobacter xylinus E25* strain, i.e., multiplying individual bacterial colonies to an amount that could be used to inoculate the appropriate volume of the culture medium. The synthesis was conducted on a laboratory scale in incubation Erlenmeyer flasks were kept at 30 °C ± 2 °C and a pH of 5.75 ± 0.03 for two days. The second stage was a stationary fermentation at controlled conditions—cultivation in trays in which the cellulose film was created on the culture medium surface. The obtained material was subjected to several purification steps: rinsing with hot water, submerging in NaOH solution at 80 °C, 1% acetic acid (until neutralization), and finally purification with water. The wet material was leveled using a mechanical press. The thickness of the obtained films was 2 mm. After washing and leveling, the samples were dried in a dryer at 45 °C to remove water more accurately. In the case of the ex-situ composites, the purified BNC film, in its wet jelly-like consistency, was further processed; the description of this process is presented in Section 2.3.The grammage (i.e., base weight, g/m^2^) was determined by means of a weight method. Details of the culture, the medium composition, final purification, and sterilization were described in earlier works [37,38]. The efficiency of BNC production was expressed as the ratio of the dry sample per medium volume (E, g/L), and the yield of the biosynthesis (Y, %) was expressed as follows [39]:(1)$$Y=\frac{m_{\mathrm{BNC}}}{m_{n}}100\%$$
where m_BNC_ is the dry sample weight of BNC, and m_n_ is the weight of the carbon source in the nutrient medium.

### 2.3. Preparation of BNC/PVA Composites

BNC/PVA composites were prepared according to three methods:In-situ—in which different concentrations of PVA solution were added to the SH medium, resulting in the formation of a mixture of glucose and PVA at ratios of 2:1, 1:1, and 1:2; the PVA concentrations in the culture medium were 1%, 2%, and 4% (*m*/*v*), respectively. The samples were marked as H-1, H-2, and H-4;Ex-situ—impregnation of pure BNC in PVA solutions of various concentrations (1%, 2%, or 4% (*m*/*v*)) and heating at 80 °C for 2 h, while constantly mixing with a magnetic stirrer. The samples were marked as I-1, I-2, and I-4;Ex-situ associated with sterilization—this method involved heating pure BNC in PVA solutions (with the same composition as the ex-situ impregnation method) in an autoclave at 121 °C, at approximately 220 kPa, for 20 min. The samples were marked as S-1, S-2, and S-4.

To evaluate the bacterial viability in the presence of PVA, the bacterial culture was placed on Petri dishes with the modified SH medium and then incubated for two days at 30 °C in three diluted solutions: 1/100, 1/1000, and 1/10,000. Two replications were performed for each dilution and the average number of bacteria was calculated. The process of BNC composite production is shown in Figure 1.

### 2.4. Analysis of BNC/PVA Composites

The structure of the BNC/PVA composites was characterized by ATR-FTIR spectroscopy using a Mattson Genesis II spectrophotometer (Madison, Wi, USA) and an ATR device, produced by MiracleTM Pike Technologies, equipped with zinc selenide crystal. Each spectrum was produced with an average of 64 scans; the resolution was 4 cm^−1^.

X-ray diffraction measurements were made using the X’Pert PRO system, applying nickel filtered CuKα radiation (wavelength 1.54056 Å) in the 2θ angle range from 2° to 40°. The degree of crystallinity (X_c_, %) was calculated as the ratio of the surface area of signals corresponding to the crystalline phase to the total surface area under the XRD pattern [40,41]:(2)$$X_{c}=\frac{A_{\mathrm{cr}}}{A_{\mathrm{cr}}+A_{\mathrm{am}}}100\%$$
where A_cr_ is the surface area under crystalline peaks, and A_am_ is the area of amorphous halo.

The sample morphology was studied by SEM and AFM using the1430 VP microscope produced by LEO Electron Microscopy Ltd. (Lewes, UK), and the MultiMode microscope (Veeco Instruments, Inc., Santa Barbara, CA, USA) equipped with a NanoScope IIIa and a Quadrex controller, respectively. Distinct areas of the sample surface were photographed at various magnifications. Additionally, roughness parameters were determined from AFM images as the arithmetic mean deviation of the registered profile (R_a_) and the root mean square (R_q_):(3)$$R_{a}=\frac{1}{N}\overset{N}{\underset{j=1}{\sum }}\left|Z_{j}\right|,\;\;\;\;\;\;\;\;R_{q}=\sqrt{\frac{1}{N}\overset{N}{\underset{j=1}{\sum }}Z_{j}^{2}}$$
where Z_j_ is the deviation of a given profile point, and N is a number of measuring points.

Moreover, R_max_, which represents the maximum roughness, i.e., the largest distance between the highest and the lowest point of the AFM image in a given scanning area, was established.

The amount of absorbed water was assessed following ISO 62:2008. The samples were thoroughly dried for 24 h at 50 °C and weighed on an analytical balance. Then, they were placed in a container with deionized water at 23 °C. After a specified sorption time (6–72 h), the samples were removed, dried with filter paper, and weighed. The absorbed water content (C, %) was calculated from the following relationship:(4)$$C=\frac{m_{2}-m_{1}}{m_{1}}100\%$$
where m_1_—the dry sample weight; m_2_—the wet sample weight.

Surface properties were determined by a contact angle (θ) measurement using a DSA G10 goniometer (Krüss GmbH, Hamburg, Germany). θ values of drops on a horizontal sample were appointed for two liquids of different polarities (glycerol and diiodomethane); hence, the surface free energy (γ_s_) was calculated based on the Young equation:γ_s_ = γ_sl_ + γ_l_cosθ(5)

The value of each contact angle was the average of 10 values.

Mechanical properties were determined by two methods. The first method was a classic tensile test performed on the Instron 1026 apparatus following the ISO 527-3:1998 standard. At least 10 measurements were made for each type of material in the hydrated form and with a standard paddle shape. Before the test, the samples were immersed in distilled water for 2 h and then dried with a filter paper. The breaking stress, strain at break, and Young’s modulus were determined from the recorded stress–strain curves.

In the second test, the bursting strength with the ball method was determined following ISO 12625-9: 2007. In contrast to the tensile tests, the dried specimens, which were in the shape of a disc with a 5 cm diameter, were used in these studies. There were at least 7 replications for each type of sample. The analyses were carried out using the MTS Criterion testing machine, model C43.104 (with a maximum nominal force of 10 kN), equipped with a clamping system and an indentation system, and a 10 mm spherical indenter. The main parameter obtained from this test was the burst index (BI), i.e., the bursting force of the sample divided by the base weight (grammage) of the conditioned sample (expressed in N·m^2^/g).

## 3. Results and Discussion

### 3.1. Preparation and Characterization of BNC/PVA Composites

Bacteria growing in an aqueous nutrient produced a film at the interface between the liquid phase and air. According to the literature, this film was mainly type I cellulose, comprising two crystallographic forms referred to as I_α_ and I_β_ [39].

The BNC and BNC/PVA pellicles, in a wet form, were jelly-like materials containing about 93–98% water. After drying, thin non-transparent membranes were formed. However, the addition of PVA caused a slight improvement in transparency.

The effect of PVA on cellulose biosynthesis was assessed based on the number of bacterial colonies formed on Schramm Hestrin agar medium after 2 days of inoculation (Table 1). The number of bacteria produced decreased with the increasing of the PVA concentration in the SH medium. At the highest 4% PVA concentration, the number of bacterial colonies was almost three times smaller than in the standard SH medium. This was due to the changes in the physical properties of the modified culture medium, e.g., the density and viscosity increased significantly with the PVA concentration, which affected the movement speed and life processes of the *Gluconacetobacter xylinus* E25 strain. The decreasing amount of bacterial colonies on the plates resulted from addition of PVA to the nutrient medium, as this bacteria strain used to live in a specific environment, i.e., the standard SH substrate.

The efficiency (E, g/L) and the biosynthesis yield (Y, %) calculated for the composites formed in-situ increased in the presence of PVA compared to those calculated for the neat BNC. The highest E and Y parameters were found for the sample with the lowest PVA concentration, and these parameter values decreased with the increasing content of PVA in the SH medium. Lower E and Y values for the composite with 4% PVA could be explained by the low bacterial population in this sample.

Another parameter showing the effect of PVA on the bacterial strain was the thickness of the formed cellulose pellicles. The thinner BNC film was formed on the medium surface at a higher concentration of the PVA solution. Moreover, the somewhat stiff and compact cellulose sheets became more delicate, elastic, and gelatinous when PVA was present in the nutrient medium.

Table 2 shows the base weight of the dry samples. The neat BNC was characterized by the most negligible base weight (2.21 g/m^2^) compared to the base weight of the composites. In the presence of PVA, the grammage increased most in the systems obtained by the ex-situ/impregnation method, and least in the samples obtained by the in-situ process. Moreover, the base weight and grammage increased with the growing PVA content when applying the ex-situ methods, whereas this trend was the opposite for the in-situ method. Thus, it could be concluded that PVA was an additional nutrient medium for bacteria as a slight increase in the base weight in the samples obtained by the in-situ method was observed, while the PVA that was added to the already produced cellulose was just a physical additive (i.e., the second component of the mixture) in the ex-situ methods. Similar results were observed by Gea et al. [42].

It was also found that the physical form of the solid samples depended on the drying method. In the samples that were dried slowly in the air and then in a vacuum dryer, relatively smooth films, with no defects and no specific structure that was visible to the naked eye, were obtained.

### 3.2. ATR-FTIR Spectroscopy

Structural characterization of the obtained composites was performed with the help of ATR-FTIR spectroscopy. The spectra of all BNC/PVA samples were similar to that registered for pure BNC (Figure 2). However, the hydroxyl band in the range of 3000–3600 cm^−1^ was wider in the composites’ spectra than that of the neat BNC, which confirmed the formation of intermolecular interactions between hydroxyl groups of PVA and BNC. This band, with a maximum at 3345 cm^−1^, was quite sharp in the BNC spectrum, while it was wider and flatter in the PVA spectrum (spectra of pure polymers not shown).

Moreover, in the infrared spectra of all BNC/PVA composites, one can distinguish a carbonyl band with a maximum at 1730/1713 cm^−1^ derived from the vibrations of the not-fully-hydrolyzed acetate groups in PVA, which is seen in Figure 2 in the enlarged part of the infrared spectra of the in-situ composites. This band was not seen in the BNC spectrum. The second band characteristic of PVA, which was not present in the BNC spectrum, was observed at 836 cm^−1^. This band was associated with the deformation vibrations of the -CH_2_ moiety in vinyl polymers [43,44]. The third band, which was also not present in the BNC spectrum, was observed at 1250 cm^−1^ and was attributed to the C-O stretching vibrations that are typical of acetate groups [43,44]. The carbonyl band, present in all spectra of the composites, indicated that the water-soluble PVA could not be washed away, despite expanding rinsing of the samples after synthesis. Thus, it was proven that the PVA molecules had been permanently embedded in the cellulose structure, either by hydrogen bonds or by physical entanglement, as schematically illustrated in Figure 3. Hydrogen bonds between the macromolecules of the same type and between BNC and PVA macromolecules could be formed in all systems.

The intensities of the characteristic bands of BNC regularly decreased as the PVA content increased in all composites. In the fingerprint range, the bands of both polymers overlapped, mainly in terms of C-O stretching vibrations and CH_2_ deformation vibrations.

### 3.3. XRD Analysis

The XRD pattern of pure BNC contained relatively intense and narrow reflections at 2θ = 14.0° and 22.6°, which could be assigned to the (100) and (110) crystallographic planes in the I_α_ cellulose triclinic cell, respectively (Figure 4). These peaks could simultaneously be attributed to (1$\bar{1}$0) and (200) in the I_β_ polymorphic form [39,42]. A low-intensity peak at 16.6° was described by the Miller indices of (010) for I_α_ or (110) for I_β_. The cellulose I_α_ prevailed over cellulose I_β_.

These signals also existed in all composites’ XRD curves, but their intensities varied slightly depending on the amount of PVA introduced (Figure 4). This means that the BNC crystallographic structure did not change in the composites.

The XRD pattern of neat poly(vinyl alcohol) revealed a signal at 2θ = 19.3° and a weak shoulder at 22.5°. These peaks were characteristic of semi-crystalline PVA and represented reflections from (10$\bar{1}$) and (200) monoclinic unit cells [45].

The PVA characteristic reflex at 2θ around 20° appeared in the XRD curves of the BNC/PVA composites obtained by the in-situ process, and it was more pronounced for the sample with the highest PVA concentrations. This indicates that bacteria did not consume a significant part of PVA macromolecules during BNC biosynthesis. However, there were no PVA signals in the samples prepared by the ex-situ/impregnation method, which was due to the total lack of PVA-ordered structure. This can be explained by the fact that the formation of thin PVA film on the cellulose nanofibers was mainly because of hydrogen bridges. Thus, in this thin poly(vinyl alcohol) coating, no regular ordering of PVA macromolecules could occur. In the samples obtained by the ex-situ/sterilization method, at lower concentrations of PVA (S-1, S-2), the reflections that are characteristic of this polymer also did not appear in the XRD patterns. Only the sample with the highest 4% concentration of PVA (S-4) showed partial PVA ordering.

The full width at a half maximum (FWHM) of the BNC signals determined at 2θ = 14.0° and 22.6° was generally smaller for all composites, and this parameter decreased slightly with increasing PVA content in the composites produced by in-situ and ex-situ/sterilization methods (Table 3), suggesting slight growth in the crystal size of BNC. In the samples obtained by the ex-situ/impregnation method, the opposite trend was observed; a growth in FWHM values with the increasing of the PVA content.

The degree of crystallinity (X_c_) determined from X-ray diffraction patterns is presented in Table 3. As can be seen, the X_c_ of the composites obtained by all methods was lower than that of BNC, and it decreased with the increasing of the PVA content; the lowest X_c_ value was found for the H-4 sample. Such results indicate that PVA made the cellulose chain ordering more difficult in the composites; thus, PVA reduced the amount of crystalline phase.

### 3.4. Morphology Studies by SEM and AFM

The SEM photographs of bacterial cellulose and its composites (Figure 5) show the typical fibrous structure of the materials, while the PVA film appears to be smooth with a few minor defects that arose during the sample preparation. The estimated fiber thickness is in the order of several dozen nanometers.

The analysis of SEM images of the composites obtained by the in-situ method indicated a good mixing of both components; thus, it was not possible to observe distinct morphological changes, nor any separate phases. The composite structure seemed more plain and homogenous (Figure 5c) as a result of hydrogen bonding between the components. In these materials, the fiber structure was tightly woven with no clear free spaces between fibrils.

The SEM images of the composites made by ex-situ methods differed from the previous ones. For these samples, the fibers were partially coated and even “glued” to PVA, which may have prevented the phase separation that is typical of immiscible systems. The impregnated samples had slightly more free spaces between fibrils. These gaps probably arose due to the intensive stirring during the sample production, which led to the partial loosening of the tangled fibers. However, the structure of the sterilized samples was relatively compact owing to the preparation method, i.e., the high temperature and higher pressure.

The AFM technique provided the images of the surface morphology of the samples and the roughness parameters that determined the ability of the adhesion of the studied sample to other substances or tissues. Figure 6a shows a height image and Appendix A shows phase and amplitude images of the neat BNC where a network of tangled fibrils was seen; these are similar to the SEM image of BNC. Figure 6b shows a cross-section illustrating the pits and hills on the sample surface, and Figure 6c presents a depth histogram of the film.

Appendix A shows 3D images of BNC and its composites obtained by the applied methods. One can see a fibrous structure of all sample surfaces in the images, which are similar.

Moreover, the fibril thickness can be measured from AFM images for a smaller scan area. The sample surfaces’ height images are shown in Figure 7a,b, where it can be seen that the thickness of a single BNC fiber was about 40 nm, and a bundle of cellulose nanofibers had a diameter of about 100 nm. In the samples obtained by the *in situ* method, the average fiber thicknesses were close to the value for BNC alone. The fiber thickness estimated from AFM and SEM images coincided. However, the dispersion of the values was large because microscopic images usually show the fiber aggregates, not single fibers. Therefore, these values were mainly of comparative significance, indicating the thickening of the BNC fibers due to PVA sticking to them. Based on the measurement of approximately 20 typical fibers, it was estimated that their thickness for the ex-situ samples increased in the range of 30–76% (on average 53%) compared to the unmodified BNC. The following image (Figure 7c) shows the places where cellulose fibers were covered with a PVA layer, which caused their thickening. The AFM images recorded with varied lever oscillations (phase images) clearly show darker and lighter areas of the fibers (Figure 7d), which confirms the presence of crystalline and disordered structures in the same fiber. Various entanglement, twisting, and branched cellulose macromolecules are visible together with disordered regions observed by other authors [46].

The roughness parameter values calculated from AFM images are listed in Table 4. The PVA film surface was very smooth in contrast to BNC, for which the R_q_ and R_a_ values were over five times higher. The surface roughness of BNC was caused by the network of the entangled nanofibrils that appeared in a semi-crystalline form. Moreover, the cellulose chains were stiffer than the PVA chains, which caused stresses and then increased the roughness parameters of the BNC.

The roughness parameter values of the composites obtained by the in-situ method were lower than those for pure BNC. Moreover, the fiber width did not change significantly compared to that of BNC. These sample surfaces were smoother than BNC alone, suggesting that the cellulose fibers were already coated by PVA during biosynthesis and stronger interactions between the components in these systems were possible. The second reason was probably partial consumption of PVA by *Gluconacetobacter xylinus* bacteria.

However, in the composites obtained in both ex-situ processes, one can observe a more pronounced influence of PVA on the surface morphology of the formed pellicles—apart from the thickening of the fibers, a significant increase in the surface roughness was also observed. The highest increase in R_q_ and R_a_ values was found for the composites obtained by ex-situ/impregnation. Furthermore, parameter R_max_, which characterizes the maximum protuberances (or concavities), was much larger for the composites obtained by the ex-situ/impregnation method than in the case of other samples.

In the ex-situ sterilized samples, high temperature (121 °C) and autoclave pressure caused loosening of the dense fibrous structure, facilitating the penetration of PVA macromolecules into the cellulose network. Then, the drying process of the composites prepared in this way caused “sticking” of the PVA-coated fibers and thereby partial smoothing of this heterogeneous surface, resulting in somewhat lower R_q_ and R_a_ values than these for the composites obtained by the ex-situ/impregnation method.

It is generally accepted that higher roughness leads to greater surface areas of biopolymer films, which is beneficial for some biomedical applications [47].

### 3.5. Surface Properties by Contact Angle Measurements

The measurements of contact angles (θ) provide information on the hydrophilic or hydrophobic nature of a sample, which is essential when the biocompatibility of the materials is assessed because it affects the adsorption of protein and its interaction with cells [48].

Based on the measurements of θ with the test liquids of different polarities, the surface free energy and its polar and dispersion components were calculated by the Wendt–Owens method, and the results are presented in Table 5.

Although OH groups existed in both polymers, the PVA sample had much lower polarity than BNC, as evidenced by lower values of γ_s_ and the polar component γ_s_^p^. The lower hydrophilicity of PVA indicated the orientation of polar hydroxyl groups to the film’s interior, where they were held through hydrogen bonds. However, in BNC film, more polar groups were present on the sample surface. Such differences in the surface polarity of both polymers resulted from the structure of these polymers. PVA macromolecules are more flexible and can more easily adjust to each other than stiffer BNC macromolecules. Similar results were obtained by AFM studies, i.e., flat PVA surfaces indicated inside polymer interactions and rough BNC surfaces suggesting difficulties in chain arrangement.

The γ_s_ values of the composites obtained by the in-situ and ex-situ/impregnation methods were higher than those for the neat polymers, while the surface free energy values of the composites made by the ex-situ/sterilization method was lower than that for BNC and higher than for PVA. Generally, in all cases, γ_s_ values increased slightly with the increase in the PVA content. The same relations were observed for the γ_s_^d^ values. At the same time, the polar component values of all composites were approximately similar to that of the BNC value. These parameter values also slightly increased with the increasing of the PVA content. This indicates that there were similar contents of hydrophilic hydroxyl groups on the surface of both BNC and the composites; PVA did not significantly influence the hydrophilicity of the obtained composites.

### 3.6. Water Absorption

Determination of the ability to absorb water is necessary for assessing the suitability of the material designed for dressing products, where it is crucial to absorb wound exudates.

The amount of water absorbed, expressed in weight percentage, is listed in Table 6. Films of BNC and its composites revealed a systematic increase in water absorption with the soaking time of the materials in deionized water. An analogous parameter could not be determined for PVA film due to its water solubility.

Pure BNC was initially characterized by moderate water uptake, but after 3 days, it was already about 300%. Much better sorption properties were exhibited by the ex-situ composites for which the amount of water absorbed in subsequent measuring periods was about two times greater or even more than that of BNC. The composites prepared in the ex-situ impregnation process exhibited the best absorption properties. The I-4 sample was unique because after 3 days of soaking, the amount of absorbed water reached about 700%. Poly(vinyl alcohol) deposited on the surface of the cellulose fibers could be responsible for such excellent water-uptake ability. This behavior was obvious due to the specific morphology of the sample. There were numerous cavities between the fibers on the sample surface, and these can be observed in the SEM and AFM images.

The materials obtained by the ex-situ sterilization method also had good absorption properties. The amount of water absorbed by S-1 and S-2 composites was similar to those for the I-1 and I-2 composites. However, the S-4 sample absorbed less water compared to the excellent I-4 sample. Probably at higher PVA content, this polymer was not only deposited on the surface of the BNC fibers; in addition, some of the polymer molecules penetrated inside the cellulose film. The PVA molecules trapped in the fibrillar areas had a limited ability to absorb water compared to the PVA molecules on the surface.

Only the composites obtained by the in-situ method exhibited a lower ability to absorb water compared to BNC (the exception was H-4 soaked for 24–48 h). This could be explained by the presence of a significant amount of PVA built into the internal structure of the cellulose matrix, where strong intermolecular interactions and physical cross-links could occur. This structure hindered the access of water to PVA molecules and caused water absorption to be lower than in the samples produced by the ex-situ method.

Moreover, the water uptake increased significantly with the increasing of the PVA content in all composites.

### 3.7. Mechanical Properties

The typical stress–strain curves of the studied samples showed an initial area of proportional dependence of stress from strain, for which the Young’s modulus was determined. While stretching a sample, the tensile force increased, leading to the fracture of the sample. From the tensile tests, the Young’s modulus, stress, and strain at break were determined.

Figure 8 shows the tensile test results of the prepared samples. A clear dependence of the results on the methods of obtaining composites and PVA content can be seen. Generally, one can notice that the stress at break and Young’s modulus decreased with the increasing of the PVA content, while the strain at break exhibited the opposite effect within the set of the samples obtained by one method.

The Young’s modulus values of the in-situ composites were similar or lower than this of BNC, whereas the stress and strain at break had the highest values among the studied samples. Such behavior of these composites indicated the highest flexibility and resistance to breaking due to the presence of PVA in BNC fibers.

The ex-situ/sterilization process in the presence of PVA resulted in a significant increase in the Young’s modulus value of the S-1 sample, whereas for the S-2 sample, this modulus increased only slightly, but for the S-4 sample, this modulus value decreased. The breaking stress value increased slightly for the S-1 sample, but this quantity decreased for the S-2 and S-4 composites. Moreover, this composite type was most sensitive to fracture as very small deformations were enough to break these samples, and the strain at break values were the lowest for these composites; thus, these composites seemed to be the stiffest and most fragile materials.

The composites obtained by the ex-situ/impregnation method exhibited lower Young’s modulus and breaking stress values and higher stress at break values compared to the neat polymers, which indicates low tensile strength and good flexibility for these materials.

Similar dependence regarding the effect of PVA on the mechanical properties of BNC materials immersed in a PVA solution was found by Qiu and Netravali [49].

Another test applied for mechanical studies was the burst tests used under the EN ISO 12625-9: 2007 recommendations to assess the resistance to the mechanical indentation. The maximum pressure force perpendicular to the surface of the cellulose sample was determined. This method is used for characterizing packaging materials and is recommended for the evaluation of materials with potential uses as implants.

In contrast to the tensile tests, the dried samples were used in this case. The results obtained for the BNC/PVA composites showed that the addition of poly(vinyl alcohol) to the culture medium reduced the burst index (BI); however, the composites obtained by the in-situ method exhibited higher values for this parameter than the composites obtained by ex-situ methods, which indicated their better resistance to bursting (Figure 9). Moreover, higher concentrations of PVA led to greater BI values in these composites, which suggested an increase in the elasticity and durability of the sample due to the presence of PVA.

The samples obtained by the ex-situ methods also exhibited a reduction in BI compared to BNC; however: this parameter decreased with the increasing of the PVA content. It seems that the ex-situ methods were not suitable for enhancing the burst index of the samples. The manner in which PVA attached to the BNC in both ex-situ methods did not affect this parameter.

It can be stated that the samples prepared with the addition of poly(vinyl alcohol) to the culture medium are good starting materials for further modifications to improve the bursting strength. A simple method that would probably positively affect this parameter is extending the culture time and enriching the culture medium with additional carbon sources.

## 4. Discussion

The latest studies in the literature show the significant interest in BNC [50] and BNC/PVA composites that can be used in various industrial sectors, particularly in medicine [32,34,51,52,53,54,55,56,57,58,59].

However, a direct comparison of the properties of the materials obtained by other researchers is challenging due to the differences in preparation conditions and composition.

For example, Gea et al. [42], who prepared biocellulose (BC)/PVA composites in-situ and by means of the impregnation method, observed a plasticizing effect of PVA and the destruction of hydrogen bonds in BC, which resulted in a decrease in the Young’s modulus, an increase in toughness, and an improvement in transparency.

The research by Millon et al. [32,51,52] concerned PVA systems with a low content of BNC fibers (<1%), for which the mechanical properties were determined in both tensile and compression tests. They found that the tested nanocomposites that had a controlled degree of anisotropy showed improved stress and good viscoelastic behavior, allowing them to be used as orthopedic or cardiovascular implants.

Leitão et al. investigated cross-linked membranes based on biocellulose and PVA [34]. In these interpenetrating networks, prepared via the immersion of BC in PVA solution, the changes in morphology during drying were found, which resulted in a deterioration of tensile strength and lower permeability. The Young modulus decreased from 3.5 GPa in pristine BC to 1 GPa in BC/PVA.

In their review, Almeida et al. [53] present the main cosmetic applications of BNC, including composites with PVA. Citing work by Chunshom et al. [54], they state that BNC has a strengthening effect in dried-state hydrogels with PVA and simultaneously improves their thermal stability up to 200 °C. The ratios of PVA to BNC in this case were 3:1, 5:1, and 10:1. A sample with a component ratio of 3:1 showed exceptional swelling properties in the water, NaCl solution, and PBS.

In another work [55], the blend of BNC and PVA obtained by the freeze–thaw method was investigated as a material to be used for the production of artificial cornea. These hydrogel materials were characterized by high water content, good transparency, and good mechanical and thermal properties. In a recent article by Wang [56], the BNC/PVA composite was also enriched with silver nanoparticles, resulting in bactericidal properties; thus, this material can be a valuable packaging material.

In the work of Ludwicka et al. [57], biomaterials based on BNC, which were designed for the packaging industry, were reviewed. In this article, particular emphasis was placed on intelligent materials for food packaging. The activity of packaging films could be achieved through the incorporation of bioactive compounds to BNC and its composites, such as antimicrobial agents, oxygen scavengers, moisture regulators, and flavor and odor absorbers.

Promising biomaterials for artificial blood vessels were developed by Tang et al. [58]. In this work, BNC tubes were modified by PVA impregnation, and new types of tubular composites were obtained using a thermally induced phase separation method. This modification led to improved mechanical properties and water permeability.

Recently, the composites of BNC/PVA with boron nitride as a filler were used to fabricate scaffolds for bone tissue engineering by means of the 3D printing method [59]. In this case, the material’s tensile strength was decreased with the addition of BC, but the viability of human osteoblast cells was increased.

Considering the tested properties of our produced composites, two types of materials are worth mentioning: the samples obtained by the in-situ method and those obtained via ex-situ/impregnation. The in-situ composites exhibited the best mechanical properties; however, they had poor water absorption ability and low surface roughness, which could limit their usefulness as wound dressings. This drawback may be removed by a longer incubation time or enrichment culture medium in an additional carbon source (e.g., glycerol, saccharose), whereas the ex-situ/impregnation composites revealed the best ability to absorb water and high surface roughness, and they were also stretchy. These properties make these materials useful in wound dressings.

Good repeatability of the composite properties was obtained owing to the maintaining of the synthesis regime on a laboratory scale. Moreover, industrial implementation, following the developed procedures, is possible at the Bowil Biotech Ltd., which has a large fermenter in which bacteria can be grown in a volume of 10–2000 L. Bacteria cultivation in this industrial fermenter allows complete control of the process conditions (temperature, pH, mixing speed, and oxygen level).

Finally, the designed method of BNC/PVA composite preparation can be considered to be an environmentally friendly method because the matrices are obtained in biosynthesis, in which no hazardous solvents and other toxic substances are used. Appropriate modification of the composites and a strictly defined biosynthesis procedure make it possible to propose potential applications of these materials in the medical or cosmetic industries, as shown in Table 7.

## 5. Conclusions

Three methods (in-situ: bacterial culture; ex-situ: impregnation and sterilization) were developed to obtain the BNC/PVA composites. The addition of PVA to the culture medium resulted in greater efficiencies and yields of the biosynthesis. Thus, it can be stated that PVA has a positive effect on the synthesis of BNC. A more significant increase in dry mass and base weight was observed for all composites as compared to the pure bacterial cellulose produced by the standard cultivation method.

The morphologies of all samples varied somewhat owing to the preparation method. A typical tangled nanofibrillar network was present in all samples, but the thickness of the fibril was greater in the composites obtained by the ex-situ methods than those obtained via the in-situ technique. In the samples obtained by the in-situ method, PVA was incorporated into the BNC fibers, while in the systems obtained by ex-situ methods, the fibers were mainly surface-coated with PVA. ATR-FTIR spectroscopy confirmed the presence of intermolecular interactions between BNC and PVA macromolecules. The degree of crystallinity of BNC was reduced in all composites, indicating the influence of PVA on the cellulose chain arrangement. However, the crystalline BNC structure existed in the produced composites, and XRD studies confirmed the presence of the I_α_ and I_β_ phases of BNC in all samples.

The BNC/PVA composites were characterized by a high ability to absorb water so that these composites could be used as dressing materials dedicated to the wounds of various etiologies. It can be concluded that the impregnation of the cellulose nanofibers by PVA ensures the highest water absorption. High water uptake by these materials allows them to be soaked with appropriate medicines, e.g., antibacterial medicines or painkillers, which ensures the high sterility and effectiveness of wound dressings. These materials can also be used as drug carriers and as a barrier to external pollution, with the ability to pass oxygen.

Moreover, the hydrophilicity of the BNC/PVA composites can be partly modified through the use of a suitable preparation method and composition.

The highest stress and strain values at break were found for the BNC/PVA composites obtained by the in-situ method. However, the samples obtained by the ex-situ/impregnation method were characterized by good stretchability.

## Figures and Tables

**Figure 1 materials-14-06340-f001:**
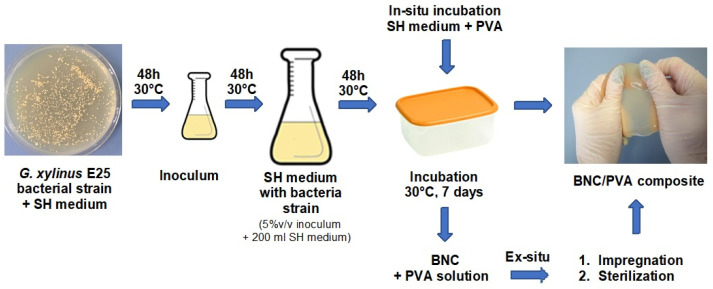
Scheme of BNC composites’ production by in-situ and ex-situ methods.

**Figure 2 materials-14-06340-f002:**
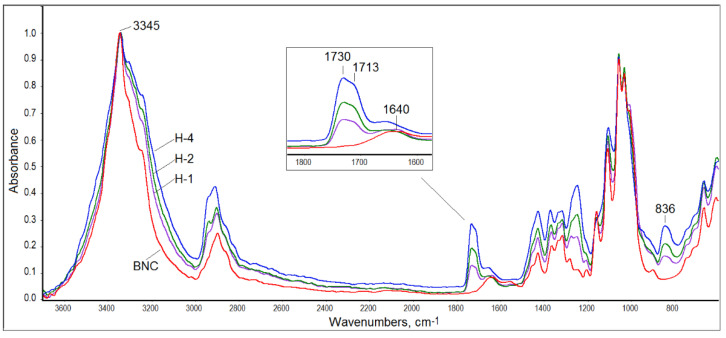
ATR-FTIR spectra of BNC and BNC/PVA composites obtained by the in-situ method (the insert shows an enlarged spectrum fragment in the carbonyl range).

**Figure 3 materials-14-06340-f003:**
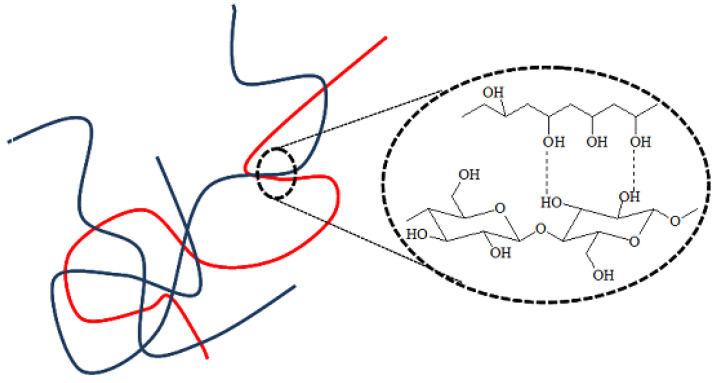
Schematic structure of BNC/PVA composite: physical cross-linking (chain entanglement, hydrogen bonds).

**Figure 4 materials-14-06340-f004:**
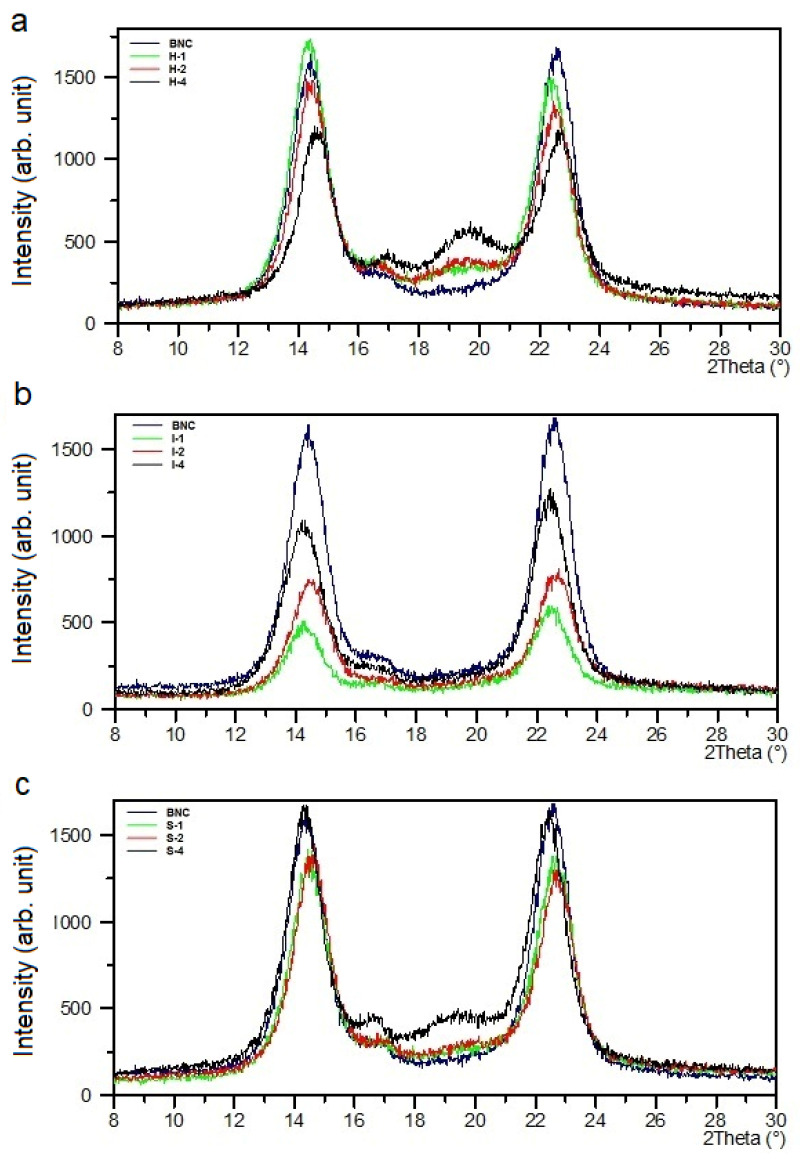
XRD patterns of BNC/PVA composites obtained by the in-situ (**a**), ex-situ/impregnation (**b**), and ex-situ/sterilization (**c**) methods.

**Figure 5 materials-14-06340-f005:**
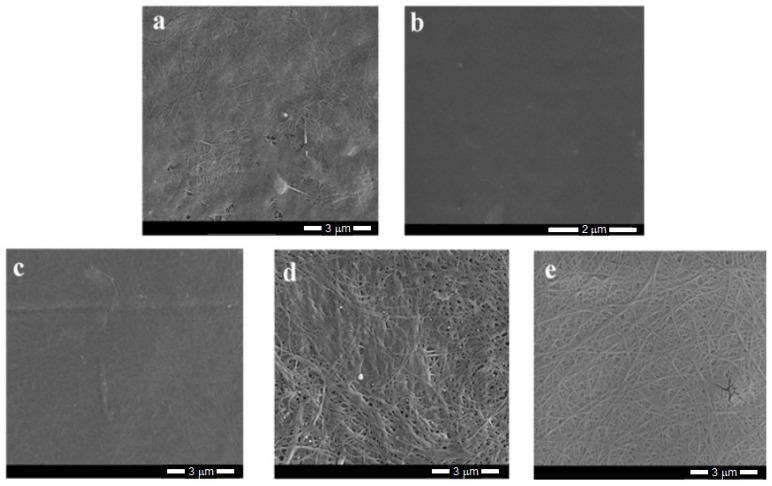
SEM images of BNC (**a**), PVA (**b**) and selected BNC/PVA composites (at 4% PVA) obtained by in-situ (**c**), ex-situ/impregnation (**d**), and ex-situ/sterilization (**e**); magnification 50,000×. The scale bar is equal to 3 μm; for PVA (picture b), it is 2 μm.

**Figure 6 materials-14-06340-f006:**
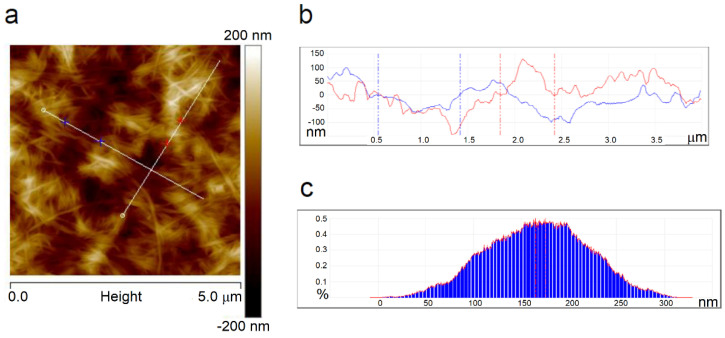
Two-dimensional-AFM images of BNC: height (**a**,**b**); cross-sections along the lines shown in Figure 6a, and depth histogram (**c**).

**Figure 7 materials-14-06340-f007:**
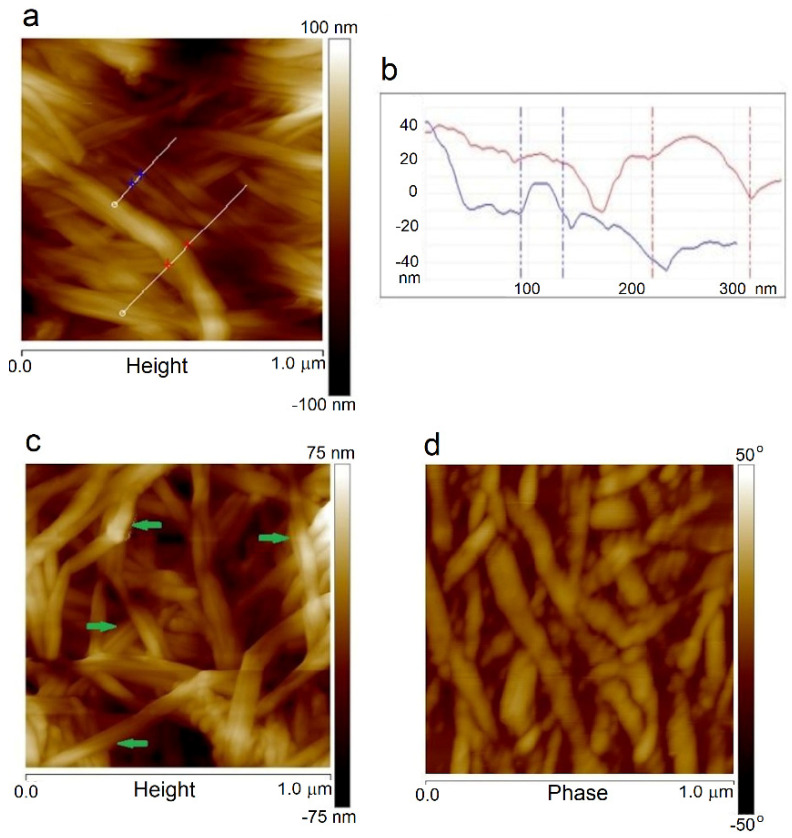
Height image of H-2 (**a**) with the fibers marked for which thickness was measured, as shown in the cross-section (**b**); height image of I-2 in which the arrows indicate the thickening of the PVA coated fibers (**c**); and phase image of S-2 showing darker and lighter parts of the fiber due to differences in order (**d**).

**Figure 8 materials-14-06340-f008:**
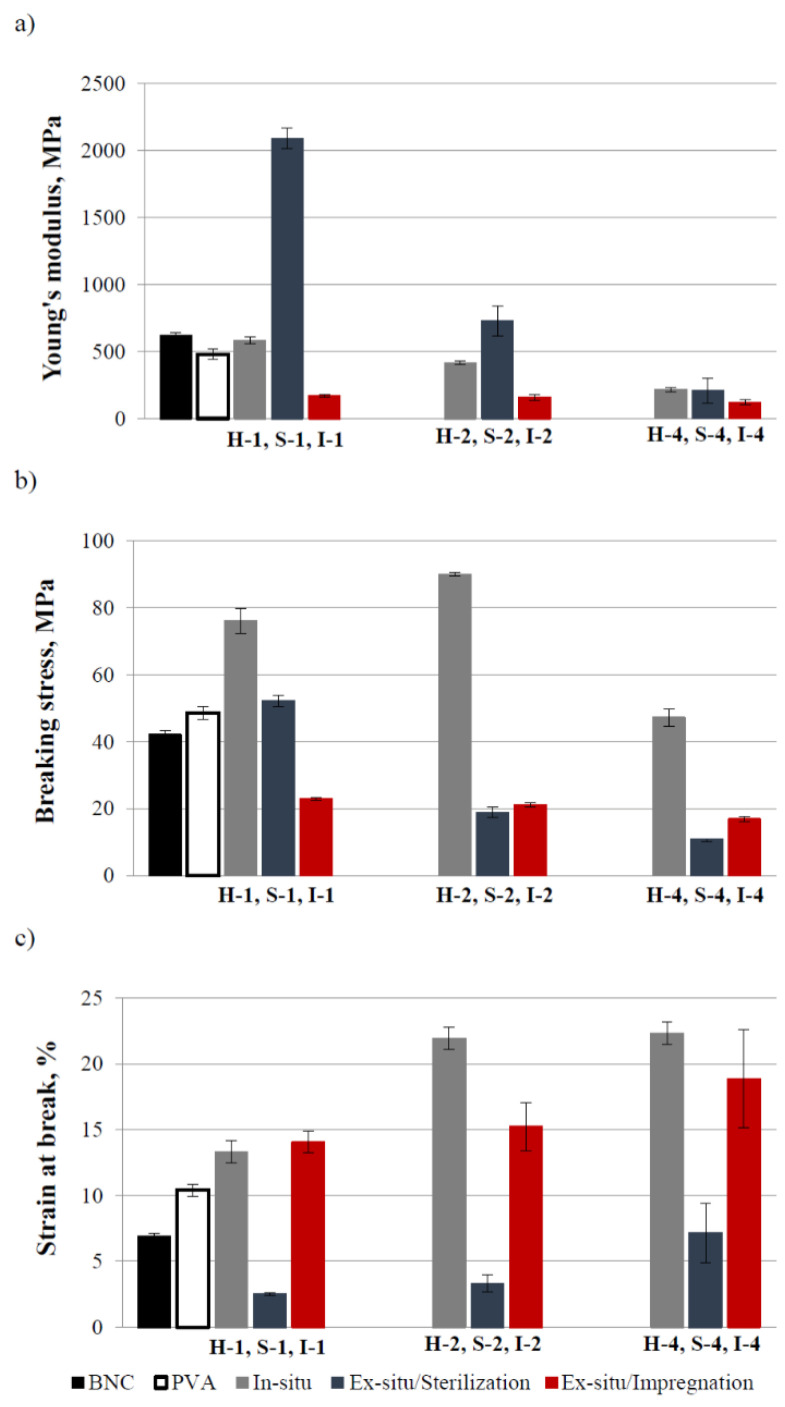
Mechanical properties of wet samples of BNC, PVA, and BNC/PVA composites obtained by different methods: Young’s modulus (**a**), breaking stress (**b**), and strain at break (**c**).

**Figure 9 materials-14-06340-f009:**
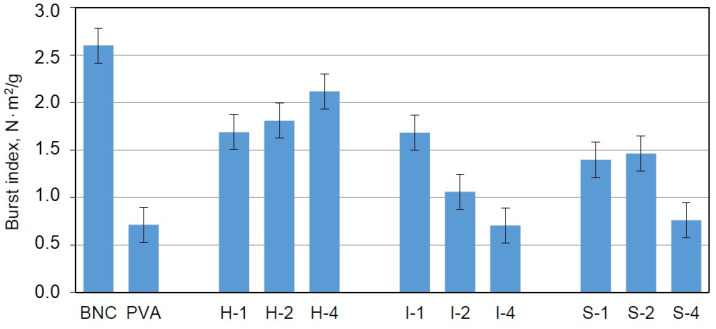
Burst index of dry samples of BNC, PVA, and BNC/PVA composites obtained by three methods.

**Table 1 materials-14-06340-t001:** A number of bacteria colonies (CFU/mL) in SH culture media (with and without the addition of 1, 2, and 4% of PVA), efficiency (E, g/L), and yield (Y, %) of biosynthesis.

Concentration of PVA, %	Number of Bacteria Colonies, CFU/mL	E, g/L	Y, %
0	1,400,000	1.507	1.795
1	1,190,000	2.291	2.729
2	1,150,000	1.892	2.254
4	490,000	1.783	2.124

**Table 2 materials-14-06340-t002:** The base weight of dry BNC and BNC/PVA composites obtained by various methods.

Sample Abbreviation	Method of Preparation	Concentration of PVA Solution, g/100 mL	Dry Weight ^1^, mg	Basic Weight of the Dry Sample, g/m^2^
BNC		0	44.2	2.21
H-1	in-situ	1	67.2	3.36
H-2		2	52.3	2.62
H-4		4	55.5	2.78
I-1	ex-situ impregnation	1	77.7	3.88
I-2		2	116.0	5.80
I-4		4	174.4	8.72
S-1	ex-situ sterilization	1	69.3	3.46
S-2		2	90.1	4.50
S-4		4	138.5	6.92

^1^ The dry weight corresponds to the 20 cm^2^ sample excised from a membrane grown in a tray onto which 550 mL of medium was poured.

**Table 3 materials-14-06340-t003:** The full width at a half maximum (FWHM, °) and the degree of crystallinity (X_c_, %) of BNC and BNC/PVA composites from XRD signals.

Sample	FWHM, ^o^, for the Signal at 2θ		X_c_, %
	14°	22°	
BNC	1.32	1.32	41
H-1	1.30	1.25	37
H-2	1.24	1.31	35
H-4	1.18	1.17	28
I-1	1.23	1.30	37
I-2	1.35	1.27	36
I-4	1.50	1.31	36
S-1	1.37	1.34	37
S-2	1.26	1.28	36
S-4	1.25	1.30	35

**Table 4 materials-14-06340-t004:** Roughness parameters (calculated for 10 µm × 10 µm scan area): R_q_, R_a_, and R_max_ of BNC, PVA, and their composites obtained by different methods.

Sample	R_q_, nm	R_a_, nm	R_max_, nm
BNC	30.7	24.3	190
PVA	5.63	4.2	48
H-1	22.8	18.2	152
H-2	26.3	20.9	187
H-4	17.4	13.6	136
I-1	72.1	55.3	642
I-2	43.0	33.5	292
I-4	59.6	46.7	345
S-1	29.8	23.6	222
S-2	36.6	28.7	247
S-4	35.3	28.2	245

**Table 5 materials-14-06340-t005:** Surface free energy (γ_s_) and its polar (γ^p^) and dispersion (γ^d^) components of the BNC, PVA, and BNC/PVA composites.

Sample	γ_s_, mJ/m^2^	γ_s_^d^, mJ/m^2^	γ_s_^p^, mJ/m^2^
BNC	43.91	28.90	15.01
PVA	33.13	25.44	7.69
H-1	42.83	29.82	13.02
H-2	44.15	29.57	14.58
H-4	47.57	31.35	16.22
I-1	47.55	30.80	16.75
I-2	48.77	33.03	15.74
I-4	49.05	32.91	16.14
S-1	35.36	21.10	14.26
S-2	37.70	22.43	15.27
S-4	39.39	23.81	15.59

**Table 6 materials-14-06340-t006:** Water absorption (wt.%) of BNC and BNC/PVA composites: C_6_, C_24_, C_48,_ and C_72_ denotes the percentage of water in the sample after soaking for 6, 24, 48, and 72 h, respectively.

Sample	Absorbed Water, Wt%			
	C_6_	C_24_	C_48_	C_72_
BNC	133	153	198	299
H-1	109	119	138	183
H-2	114	128	179	207
H-4	121	159	223	269
I-1	211	238	377	414
I-2	297	343	435	487
I-4	459	515	641	692
S-1	208	216	301	398
S-2	230	351	440	482
S-4	270	447	498	548

**Table 7 materials-14-06340-t007:** Proposed application of the composites depending on the preparation method and the composite properties.

Feature	Preparation Method		
	In-Situ	Ex-Situ/Impregnation	Ex-Situ/Sterilization
water absorption	implant	dressing for exudative wounds	wound dressing with little exudate
morphology	dressing	cosmetic industry	dressing
mechanical properties	implant	cosmetic industry	cosmetic industry

## Data Availability

Not applicable.

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
