# Peer review of "Bionanocellulose/Poly(Vinyl Alcohol) Composites Produced by In-Situ Method and Ex-Situ/Impregnation or Sterilization Methods"

_materials, 2021, doi:10.3390/ma14216340_

Round 1

Reviewer 1 Report

In the paper the authors described the characterization of the structure, morphology, crystallinity, mechanical properties of BNC/PVA composites obtained by three different ways (in-situ method and ex-situ methods combined with sterilization or impregnation) applying different techniques.

The paper is well written and the results are interesting. In my opinion the manuscript can be accepted after minor revision. I suggest the authors to review:

  • The authors should include, in the reference part and the discussion part, their previous publications papers entitled: Characterisation of Composites of Bacterial Cellulose and Poly(vinyl alcohol) Obtained by Different Methods (DÅ‚uga A, Kaczmarek H. Characterisation of Composites of Bacterial Cellulose and Poly(vinyl alcohol) Obtained by Different Methods. Fibres with Spores Incorporated. FIBRES & TEXTILES in Eastern Europe 2014; 22, 6(108): 69-74) where is described the details the obtaining of the BNC/PVA composites and the initial characterization of this compound. The authors should discuss the novelty/difference of the techniques previously reported and those presented in this new manuscript.

  • In the abstract part the authors informed that the paper aimed to obtain composites based on bionanocellulose (BNC) and poly(vinyl alcohol) (PVA) for specific biomedical and cosmetic applications; in the introduction part the authors described that the purpose of the paper was to obtain and characterize the composite, made of bionanocellulose and poly(vinyl alcohol), designed for biomedical and cosmetic applications. However, in the manuscript (results, discussion, conclusion) the authors did not mention/discuss or show the application. So, the authors must include in the paper the application of the BNC/PVA composites in order to complete it and to support the aim of the manuscript.

Author Response

As the Reviewer suggested, we cited the previous paper:

  1. DÅ‚uga, A.; Kaczmarek, H. Characterisation of Composites of Bacterial Cellulose and Poly(vinyl alcohol) Obtained by Different Methods. Fibres Text. East. Eur, 2014, 22(6), 69-74

Regarding tests related to applications of bionanocellulose and its composites, we did not conduct application tests. At this moment, we cannot conduct any experiments because the Ph.D. student, who worked on this topic, has finished her Ph.D. and left the University, and we do not have any additional samples for studies.

However, bionanocellulose produced by Bowil Biotech Sp. z.o.o. was tested in applications as bioimplant in the circulatory system, which was described in paper [38]. The studied material appeared to be suitable as bioimplant.

Moreover, Bowil Biotech produces materials such as masks used as wound dressings, face and eye masks. More information can be found on the website https://bowil.pl/en/products/

38. Kołaczkowska, M.; Siondalski, P.; Kowalik, M.; Pęksa, R.; Długa, A.; Zając, W.; Dederko, P.; Kołodziejska, I.; Malinowska-Pańczyk, E.; Sinkiewicz, I.; Staroszczyk, H.; Śliwińska, A.; Stanisławska, A.; Szkodo, M.; Pałczyńska, P.; Jabłoński, G.; Borman, A.; Wilczek, P. Assessment of the Usefulness of Bacterial Cellulose Produced by Gluconacetobacter. Mater. Sci. Eng., C. 2019, 97, 302-312

Reviewer 2 Report

The authors presented composites based on bionanocellulose (BNC) and poly(vinyl alcohol) (PVA). Samples from three different synthesis techniques were analysed for their morphological, mechanical, structural, and surface properties. Some minor comments are given as follows:

  1. Figure 2 – there is also a new band at approx. 1250 cm-1. Apparently, it is also associated with PVA. Do you have any explanation for it?
  2. Figure 4 - As the intensity is arbitrary, the units (arb. units) should be added
  3. Figure 5 – thicker scale bars, as the current ones cannot be seen. Also, the SEM images are too small.
  4. Figure 6 and 7 – I would recommend that the phase, amplitude and 3D images are moved to the supplementary files, as the fibrous nature can be clearly seen from figure 6a. In the current state, the pictures and the labels are too small.
  5. A short discussion section (comparing the results with previous studies) is missing and would improve the manuscript.

Author Response

  1. An explanation about the band at 1250 cm-1 in infrared spectra of the BNC with PVA was added:

The third band, not present in the BNC spectrum too, was observed at 1250 cm-1 and was attributed to the C-O stretching vibrations, typical of acetate groups [43,44].

  1. Hossain, U.H.; Seidl, T.; Ensinger, W. Combined in situ infrared and mass spectrometric analysis of high-energy heavy ion induced degradation of polyvinyl polymers. Polym. Chem., 2014, 5, 1001
  2. Szafran, M.; Dega-Szafran, Z. Determination of organic compounds’ structure with spectroscopic methods (in Polish), PWN, Poland, Warsaw, 1998.
  3. Fig 4 – we added “arb. unit”
  4. Fig. 5 was improved.
  5. Figs. 6bc and 7 were moved to the supplementary material (Figs. S1 and S2).

5. A short discussion section was added. This section was composed of new material and some fragments taken from the conclusion. Thus, we rebuilt the conclusions.

Reviewer 3 Report

The manuscript materials-1425939 presents some serious work, albeit not ground-breaking, on different blends of PVA and bacterial cellulose. Overall, the manuscript and the experimental work involved are of high quality, the text is well-written, the background is acknowledged, and results are presented in a clear way. The most urgent issue, to my judgment, is the covalent bond hypothesis. Even as hypothesis, readers that have studied transesterification with cellulose as the alcohol, on purpose, will surely feel disappointed to see that such hypothesis is mentioned to explain the results but never tested.

Covalent bond formation by transesterification between the hydroxyl groups of cellulose and the non-fully hydrolyzed parts of PVA is unlikely under the conditions reported, i.e., aqueous medium and no catalyst. Furthermore, I do not find any result along this paper that could not be explained by intermolecular interactions and entanglement of polymer chains.

1. Abstract: The last sentence is rather obvious and the whole abstract is too qualitative. I suggest i) removing the last sentence, and ii) adding numbers to the qualitative remarks.

2. Line 30: Arguable statement. What about cotton, for instance, which has been directly processed for ~7,000 years? I guess that the word is not "suitable" but "versatile".

3. Line 61: This paragraph fails to encompass algal cellulose, and it would be a misnomer to call algae "vegetable". Even certain animals (tunicates) produce cellulose, but that is less important. Still, I understand that the authors wanted to prepare readers for the following immersion into the different supramolecular characteristics of BNC. Thus, I suggest not to delete this paragraph, but to rephrase it.

4. Conditions of composite formation (section 2.3) might lack some detail. Whenever no temperature is specified (in situ), should we assume room T? As for ex situ processes, what was the consistency of BNC? I mean, to what extent had it been vacuum-dried or air-dried?

Also, "a different concentration" -> "different concentrations"

5. Line 158: While this method to estimate crystallinity is familiar and well-known, at least one precedent article doing this (it will be easy to find) should be cited. After all, no method for estimating the crystallinity of cellulose is "standard".

6. Line 266: "CH" -> ">CH-" (to distinguish it from "=CH-")

7. Here is where I make my point of reconsidering the covalent bond hypothesis. All results considered, everything could be explained by strong intermolecular interactions and entanglement of polymer chains, actually. 

Line 269: "either by covalent bonds or by physical entanglement

Line 301: "mainly because of hydrogen bridges and random covalent bonds"

Line 479: "where strong intermolecular interactions and covalent bonds or
cross-links could occur"

Is chemical crosslinking actually possible by the in-situ process? Non-catalyzed transesterification between the -OH of cellulose and acetate groups, without catalyst, requires absence of water and certain temperature. In water, an alkali like K2CO3, ammonia, or even NaOH would be necessary (and not just a pinch).

Carbonyl carbons, which are confirmed by FTIR, would also be there in the case of physical mixtures. Chemical crosslinking, then, could have to be tested by other ways, preferentially by solid state 13C NMR. That, or testing for acetic acid (a product of transesterification) before and after blending. If the authors cannot prove it, they should remove the statements cited above and also Fig. 3b, so as not to mislead readers.

8. XRD discussion: The pattern of BNC resembles more an idealized pattern for cellulose I-alpha that that of cellulose I-beta. Peaks at 14.0° and 22.6°, especially. In cellulose Ib, (11Ì…0) and (200) planes tend to be manifested at ~15.0 and ~23.0, respectively. Although only in qualitative terms, the authors could state that cellulose Ia prevails over cellulose Ib.

9. Finally, although the manuscript is generally well-written and mistakes are seldom found, proofreading is necessary. For instance, "referred to Ia and Ib" -> "referred to as Ia and Ib"

Author Response

  1. The last sentence in the abstract was removed. However, we were unsure what to do with the following remark “Adding numbers to qualitative remarks”. We did not insert specific numerical data because there were too many different results depending not only on the method of obtaining but also on the ratio of components, which would increase the volume of the abstract, and thus its lower transparency.
  2. Line 30; The word ”suitable” was replaced with “versatile”.
  3. Line 61; The paragraph was enriched in information on algal cellulose.

Cellulose can come from a variety of sources. The most common type is vegetable cellulose, e.g., from wood or cotton. The other type is algal cellulose present in the cell wall of algae. Due to the size and structure, the algae can be divided into microalgae and macroalgae [17,18].

  1. Zanchetta, E.; Damergi, E.; Patel, B.; Borgmeyer, T.; Pick, H.; Pulgarin, A.; Ludwig, Ch. Algal Cellulose, Production and Potential Use in Plastics: Challenges and opportunities. Algal Research, 2021, 56, 102288.
  2. John, R.P.; Anisha, G.S.; Nampoothiri, K.M.; Pandey, A. Micro and Macroalgal Biomass: A Renewable Source for Bioethanol. Bioresour. Technol., 2011, 102, 186–193.
  3. The information on temperature during BNC production can be found in Section 2.2. BNC synthesis where it is written that the temperature of the synthesis was 30oC. The production of the BNC composites was also presented in Fig. 1. Moreover, we cited two previous works where the synthesis conditions were more detailed described. We added in the text that the produced raw BNC film had jelly-like consistency, and after purification, still in the wet form, the BNC film was taken to prepare the composites by the ex-situ methods. The final BNC and the composites films were dried in a dryer at 45 oC. This information was placed in the publication. Furthermore, all these films were also freeze-dried, but these structures were not the subject of this paper.

“a different concentration” was replaced with “different concentrations”

  1. Line 158 (now 163): We cited the articles in which the degree of cellulose crystallinity was calculated according to the formulae used by us.
  2. Ogunjobi, J.K.; Balogun, O.M. Isolation, modification and characterisation of cellulose from wild Dioscorea bulbifera. Scientific Reports, 2021, 11, 1025.
  3. Rambo, M.K.D.; Ferreira, M.M.C. Determination of Cellulose Crystallinity of Banana Residues Using Near Infrared Spectroscopy and Multivariate Analysis. J. Braz. Chem. Soc., 2015, 26, 1491-1499.
  4. Line 266 (now 271): considerations concern –CH2 groups, which was corrected in the text, and also publications were cited.
  5. Hossain, U.H.; Seidl, T.; Ensinger, W. Combined in situ infrared and mass spectrometric analysis of high-energy heavy ion induced degradation of polyvinyl polymers. Polym. Chem., 2014, 5, 1001
  6. Szafran, M.; Dega-Szafran, Z. Determination of organic compounds’ structure with spectroscopic methods (in Polish), PWN, Poland, Warsaw, 1998.
  7. We abandoned the hypothesis of covalent bonds. Generally, the transesterification hypothesis was removed as during the obtaining the BNC composites, there was no catalyst and the temperature was 30oC. Thus, as the Reviewer rightly mentioned, the transesterification process would be unlikely under these conditions.

line 269 (now 277): instead of “either by covalent bonds or physical entanglement,” is “either by hydrogen bonds or physical entanglement”

line 301 (now 313): “random covalent bonds” was removed from the sentence “ mainly because of hydrogen bridges and random covalent bonds”

Line 479 (now493): “covalent bonds” was removed from the sentence, and now it looks like “where strong intermolecular interactions and physical cross-links could occur”.

We did not conduct the tests proofing chemical crosslinking, such as investigating acetic acid presence after reaction or registering solid 13C NMR; thus, as it was suggested, we also removed Fig. 3b.

  1. XRD discussion. We added information that cellulose Iα prevails over cellulose Iβ.
  2. We checked the manuscript in terms of language once again.

Round 2

Reviewer 3 Report

All issues raised by the reviewers have been addressed with commitment and responsibility. I acknowledge that experiments could be easily replicated by other research teams now, thanks to the experimental details that have been added in the revised version.